# Hesed in Ruth: A Frail Moral Tool in an Inflexible Social Structure

**Gili Kugler * and Ohad Magori ***

Department of Jewish History and Biblical Studies, University of Haifa, Haifa 3498838, Israel
* Correspondence: gkugler@univ.haifa.ac.il (G.K.); ohadmagori@gmail.com (O.M.)

**Abstract:** Scholars have paid much attention to the attribute of hesed in the book of Ruth, pinpointing it as a pivotal feature and the main message of the book. However, the protagonists in the tale do not seem to exhibit hesed out of free will or as part of their natural conduct. They rather resort to such a maneuver in order to survive and extricate themselves from dire predicaments. This article argues that the virtue of hesed attributed to the protagonists in the book of Ruth reflects a mechanism for surviving in the confining communal structure of the Judean patriarchal society, which allowed limited social mobility. While the actions exhibited in the story can be argued to be an amendment of the previous generations' perversions, the story effectively accepts and preserves the common inflexible social system.

**Keywords:** book of Ruth; divine intervention; endogamy; foreign women; hesed; levirate marriage; patriarchal society; poverty; social mobility; social perversions; social rigidness; widowhood

## 1. An Overturn in the Book of Ruth

A patriarchal and melancholic atmosphere shrouds the beginning of the book of Ruth. The plot is set in the times of the Judges, known as a period in which "every man did what was right in his own eyes" (Judg 21:25b). Famine has spread throughout the land of the Israelites and the community is in discord. Cramped into the first five verses, the exposition tells of a family of Ephrathites from Bethlehem: Elimelech, Naomi, and their two sons Mahlon and Kilion, travelling to Moab (whether due to the famine is unstated, yet implied). Such a journey can be considered dangerous and irrational, given the plot's timeframe, in which Moab is regarded as enemy territory (cf., Judg 3:12–14).[1] The exposition quickly tells of Elimelech's demise and of the death of the sons shortly after. Naomi and her two daughters-in-law are left behind, childless widows.

The tragedies that commence the plot turns the patriarchal account into a feminine tale. Indeed, the closure of the full narrative recoils the story back in the patriarchal realm, as it provides the lineage for the house of David, with no women mentioned in it (Ruth 4:18–22). However, the core narrative focuses on two of the three widows and highlights their survival in what may be considered "Job-like" circumstances.[2]

The ominous tone of the introduction takes an optimistic turn when Naomi hears that "the Lord had considered his people and given them food" (1:6b). Upon hearing this, Naomi embarks on a journey back to her Judean homeland (1:7), accompanied at the outset by her two daughters-in-law. However, Naomi endeavors to convince her daughters-in-law to turn back to Moab, to be with their families (1:8), explaining that she sees no possibility of providing them with husbands. Orpah decides to depart, and Ruth "clings" to her mother-in-law (1:14). Ruth's resolution to stay at Naomi's side, in a situation that leaves little promise for her own future, is considered uncustomary. Naomi tries to convince Ruth to join Orpah, designated twice as יבמתך (v. 15).[3]

Two childless and seemingly destitute widows set forth on a journey. Unlike stories of the ancestors returning from a sojourn to a foreign prosperous land, back to the impoverished Canaan, bearing gifts and wealth (Gen 12:16; 42:25, 43:2, cf., 32:6), in the case of the

widows, no indication is given to property, livestock, or personal belongings. Naomi depicts her state upon her return to Bethlehem as empty and divested: "I went away full, but the Lord has brought me back empty; why call me Naomi when the Lord has dealt harshly with me, and the Almighty has brought calamity upon me?" (1:21).[4] Naomi hold's God responsible for her calamity, with a tone more accusatory than Job's, who experiences a succession of tragedies, as a result of God and Satan's experiment (cf. " . . . The Lord gave, and the Lord has taken away; blessed be the name of the Lord," Job 1:21). Not only does Naomi refuse to embrace her tragic destiny like Job, but she also ignores the value of Ruth's presence in her life, a presence which would later rectify her dire situation.[5]

However, from this moment onwards, the tragic tone of the exposition fades away, launching the widows' salvation. As the *breadless* Bethlehem ("House of Bread") returns to fertility, fertility will also be bestowed upon the heirless female. This type of redemption is associated with the female concern of those times—the ability to produce an offspring. What enables this turn of events? Scholars throughout the years have proposed that the protagonists' destiny changes due to the prevalence of deeds of ḥese̱d (חסד), a term that embodies a wide range of positive attributes, such as love, mercy, grace, kindness, benevolence, and faithfulness.[6] Due to the loaded sense of this notion, we will use it here in its transliterated form, hesed.

However, this article will argue that the scholarly emphasis on hesed has illuminated only a particular marginal aspect of the book, in order to teach desired moral virtues and "Jewish" values. The widely accepted focus on hesed as the objective of the book of Ruth has diverted the attention from the harsh reality exhibited in the book, which had mandated the utilization of this attribute as a tool for surviving. The article will argue that any recognition of hesed in Ruth should go hand in hand with comprehending the social constraints and restrictions in which the protagonists lived and functioned.

## 2. The Need to Focus on Hesed

The understanding and conclusion that the book is a tale of hesed is rooted in the early days of rabbinic teachings, after the fall of the Judean community. The Israelite nation was exiled and dispersed, lacking land, king, temple, and priesthood. A dire need for establishing laws and limitations arose in order to hold together the scattered remnants of the Judean society, existing as an extra-territorial faith-based community, allowing rabbinic laws and guidance to become a dominant authority. The Tannaic rabbi Simon the Righteous is quoted as saying, "The world rests upon three things: Torah, service to God, and bestowing kindness (גמילות חסדים)."[7] The fifth century CE rabbi Zaira explains, "The book of Ruth contains neither laws of purity and impurity, nor precepts of forbidden and permitted. Then why was it written? Because of loving kindness; to teach how greatly God rewards those who do kindness (Hebrew חסדים, plural form of hesed)."[8] As with the chicken and the egg, it should be asked whether the rabbis extracted their laws and ethical guidelines from the biblical texts, or rather, after formulating these laws, searched for their validation in the scriptures for sanctioning their decisions.

## 3. Hesed in the Book of Ruth

The book of Ruth is positioned in the Jewish canon within the five books of the Megillot, four of which feature a female voice, an unconventional voice in biblical literature.[9] The book recounts a "rags to riches" tale of two marginal women. The transformation in the protagonists' situation has been interpreted since the days of the Jewish sages as an outcome of the protagonists' positive conduct throughout their experience. Similarly, hesed has been deeply rooted as the main theme of the book according to modern scholars, as recently stated by Ilana Pardes: "Reading this ancient tale allows us to rethink our current inclinations. In its intricate endorsement of hesed and perseverance in impossible conditions, Ruth's tale teaches us the value of devoting attention to these admirable human potentialities."[10]

The argument in favor of hesed as a main theme of the book finds its evidence in three trajectories in the story: (a) the relationship between Ruth and Naomi, as Ruth relinquishes everything she has known to care for her mother-in-law; (b) the interaction of Boaz and Ruth, who mutually recognize each other's hesed and fulfill each other's needs; (c) God's provision to his people, by ending the famine and providing a child, thereby allowing the family line of Elimelech to continue, culminating in the birth of David, the king of Israel.

However, on the whole, only three words in the book include the root *h-s-d* (חסד). It is hard not to assume that conclusions regarding the centrality of hesed are mostly derived by scholars' reading into the lines of the story. This article will argue that the manifestations of hesed depicted in the story are not selfless, without any expectation of retribution. They are not sublime, but rather constitute the protagonists' method to extricate themselves from harsh and fearful circumstances of a restrictive and exclusive society.

## 4. Reexamining Manifestations of Hesed: Naomi, Orpah, and Ruth

The first instance of hesed is mentioned in reference to God's potential actions. As Naomi acknowledges that fate has robbed her of motherhood, she pleads to her daughters-in-law, "Go back each of you to your mother's house. May the Lord deal kindly with you (Hebrew hesed), as you have dealt with the dead and with me" (1:8).[11] This occurrence of hesed, however, is not one of action, but of words. Naomi wishes that God brings upon Ruth and Orpah the same kindness they had shown towards her and her sons. Moreover, apart from a recognition of their kind treatment towards her, Naomi's blessing may also be a way of distancing herself from them, as they are foreigners and worshipers of other god(s), as she later states: "Look, your sister-in-law has gone back to her people and to her god(s)" (1:15a). According to Nolan-Fewell and Gunn, it is possible that Naomi's verbal generosity is no more than a polite rhetoric, while hiding her unease, resentment, and frustration due to the presence of her daughters-in-law at her side (Nolan-Fewell and Gunn 1988, pp. 103–4). It is also possible that Naomi knows of her land in Judah and seeks to reclaim it according to certain laws and traditions, seeing her daughters-in-law as a pebble in her sandal en route to getting back on her feet. As such, she urges them to return to Moab, not once but numerous times, focusing on the daughter's marital potential in the absence of wealth, land, or inheritance.[12]

As for the daughters-in-law, they find themselves at a crossroad. If they stay in Moab, Naomi returns alone and desolate, and loses the little family she had. If they join Naomi, they renounce the hope of the life Naomi's god might have given them (1:8) (see Bush 1996, p. 85). Orpah parts ways, opting for her community and faith,[13] while Ruth supposedly dismisses the blessing designated to her by Naomi's god. However, Ruth takes an oath towards this god and accepts him as her deity: " . . . Your people shall be my people and your God my God . . . May the Lord do thus and so to me, and more as well . . . " (1:16–17).[14]

Bush sees Ruth's choice and determination as a manifestation of hesed, as she casts aside concerns for her future and security, breaking all past bonds (Bush 1996, p. 87). Trible argues that such a radical commitment can only be equated to that of Abraham's.[15] However, it is debated whether Ruth truly converted and accepted Naomi's god as her only god.[16] If she accepted Naomi's "people and god," why did the author insist on continuously defining her as a Moabite, external to the community? (Whereas God is mentioned only four times in the book, the designation *Ruth the Moabite* is mentioned eight times in the short narrative). Moreover, one may ask, did Ruth not already accept God when she married Mahlon? It seems that the author prioritized dealing with the subject of foreignness.

Whereas supporters of hesed's prominence view Ruth's decision and behavior as the embodiment of kindness and selfless love for Naomi, we ask whether it is possible that Ruth realizes that no prosperity awaited her back in the land of Moab? She was already invested in Naomi and her family, and maybe looked upon negatively by her nation for marrying a Judean, as animosity runs both ways. Upon hearing of the abundance in Bethlehem, and knowing of Naomi's familial ties, she may have opted for a better chance of

altering her destiny in the land of the Israelites. As the book indicates, Ruth does not lack initiative, and will find means to modify her social predicament.

## 5. Reexamining Manifestations of Hesed: Boaz and Ruth

A new and vital protagonist is introduced in the second chapter of the book—Boaz, a kinsman of Elimelech.[17] Consecutively, we perceive Ruth's scheme, as she wishes to " . . . go to the field and glean among the ears of grain behind someone in whose sight I may find favor [אמצא חן בעיניו]" (2:2).[18] Sasson argues that Ruth implicitly refers to Boaz, who has been mentioned in the preceding verse. Since Boaz is related to Elimelech and could be a potential redeemer of the family's land, Ruth deliberately sets out from the start to meet Boaz and gain his attention and favor.[19] Accordingly, this is not a wish to achieve amity, but rather to find a benefactor in order to maintain herself and her elderly mother-in-law, thereby avoiding starvation and vehemence.

Despite the abovementioned juxtaposition, it is notable that the narrator makes clear that Ruth gleaned in Boaz's field totally by chance (ויקר מקרה, 2:3b).[20] Legal statements in the Pentateuch (Lev 19:9–10; 23:22; Deut 24:19–21) supposedly indicate that as an indigent (i.e., foreigner and a widow), Ruth is permitted to glean in the harvest fields. Nonetheless, Hubbard considers her actions courageous, since they could lead to ostracism or to physical abuse, on account of her gender, social status, and foreign ethnicity (Hubbard 1988, p. 73).

Boaz arrives at the field, blesses his servants, and notices Ruth's presence. From the overseer's explanation, we learn that Boaz is acquainted with Naomi (2:6), even though years have passed since she departed the region, and nowhere do we hear that he and Naomi made contact. Ruth is unknown to Boaz. She is labelled "a young Moabite woman" (v. 6).[21] However, Boaz calls her "my daughter," and urges her to remain among his other female servants (נערתי, v. 8), assuring her that no harm will befall her.

Ruth acknowledges Boaz's kindheartedness and thanks him (2:10). Even though it is conveyed in the book that she has embraced the Israelite god, Ruth does not thank God for her good fortune in making Boaz's acquaintance, and never acknowledges God's role in her life. This may be indicative of the authors' view that Ruth's adoption of the Israelite god was a means for joining the congregation, along with her "hesedic" behavior.

As for Boaz, he appreciates both Ruth and God's kindness. He does not yet use the word hesed but recognizes Ruth's benevolence towards her mother-in-law and her personal sacrifice (2:11–12). It is not mentioned when and where Boaz came upon this information about Ruth, but his statement clarifies why he chooses to be kind to her. Furthermore, this may reveal something about the protagonists' performance; while it may be seen as a matter of coincidence, or as God's hand directing Ruth "by chance" to Boaz's field, it may also be viewed as a fulfillment of Ruth and Naomi's collusion that ensured that Boaz hears of Ruth's wonderful nature. Sasson emphasizes that with an exaggerated gesture of humility, reserved for kings and gods, Ruth gently conveys her expectations (Sasson 1989, p. 324). She responds to Boaz by expressing gratitude, but she also persuasively asks to be treated differently from the other servants (2:13).[22]

Boaz's hospitality is persistent as he invites her to eat and ensures that she gleans as much as she desires. He even extracts stalks from bundles for her (2:16). It seems that neither Ruth's foreign identity nor her social class diminish Boaz's compassion towards her. It is debated whether this indicates Ruth's successful acceptance into his clan.[23] On the other hand, Boaz does not express a desire to court her, but only a will that she becomes one of his servants. Although acts of consideration are recognized, Boaz's actions cannot be defined as ultimate hesed, for he has something to gain—another maidservant of good nature.[24]

The second time Boaz comments on Ruth's selflessness is in the scene of the threshing floor, where he defines Ruth's hesed here as better than her first, as she has " . . . not gone after young men, whether poor or rich'" (3:10), alluding both to his advanced age and her youthfulness and beauty. Boaz tells Ruth not to fear, for he will comply, but he also discloses to Ruth the existence of another גאל, who retains first rights and might precede

Boaz in this role. In case he refuses, Boaz will act as her גאל (3:13). As the story unfolds, we see that Boaz, a prominent and wealthy person, does not miss the financial gain that entails the redemption of Ruth. Boaz's public declaration emphasizes first his acquiring the land and inheritance from Naomi (4:9) and only ipso facto acquiring Ruth (v. 10). Boaz's hesed is not a mere act done without expectation of benefit. Against the popular interpretation, we argue that like the other protagonists, Boaz also seeks possible reciprocation and gain through the conduct of hesed.

### 6. Boaz and Naomi Pick the Fruits of Ruth's Hesed

The final chapter of the book recounts the realization of Ruth and Boaz's bond. Boaz outplays the other kinsman, labelled פלני אלמני (meaning "so and so," or "someone"),[25] after he raises the stakes by introducing Ruth. Davies shows that Boaz combines two cultural customs: redeeming a relative's property, which does not require levirate marriage; and levirate marriage, which is combined with a redemption of the land bequeathed by the late husband (until a child is born and becomes the legal heir) (Davies 1983, pp. 231, 234). The kinsman may have thought that he would have to marry Naomi, who is too old to have children, or that upon marrying, Ruth he may lose his inheritance to her child. By misinforming the kinsman, Boaz's ambition is revealed: more than conducting hesed for the sake of Ruth, he is interested in the financial gain that comes along with the union.

Boaz and Ruth marry and bear a child. The narrator finds it crucial to emphasize that the conception is God's blessing, implying a divine involvement in the creation of the lineage which will bring forth David.[26] However, the child must be seen in a wider view—it is the security sought for Ruth, and moreover, for Naomi. Even though Ruth is the mother, the neighboring women congratulate Naomi for having a "next-of-kin" (גאל, a redeemer), and announce "may his name be renowned in Israel!" (4:14). The Hebrew wording of the praising (ויקרא שמו בישראל) leaves it unclear as to whose name will be renowned: God, Boaz, or the child.[27] The LXX solves the problem by offering another option: καὶ καλέσαι τὸ ὄνομά σου (your name) ἐν Ισραηλ. The renowned name will be Naomi's! This version is more suiting in the context, since the female neighbors later declare that "a son has been born to Naomi" (4:17a), and seem to name the child themselves (4:17b).[28] Scholars have debated the meaning of Naomi's role as a "nurse" (אמנת, v. 16), asking whether she is the child's guardian or an adoptive parent, or this is only an expression of love.[29] From a literal aspect, Ruth, the foreign maiden–servant–wife, fades into the shadows after bearing the child. The closing scene tells no more about her, and her destiny remains a mystery. Even though she is now married to the prominent Boaz, she is still labelled as a "maid/servant" (הנערה הזאת, 4:12). As a Moabite who "fulfilled her purpose," Ruth clears center stage for Naomi. Furthermore, in light of this conclusion, Naomi's previous acts may also be interpreted as a utilization of Ruth.

### 7. Ruth's Performance in Naomi's Shadow

Naomi is the focus of the first scenes of the plot. She is mentioned as arising, returning, hearing, and setting forth—all verbs are put in the female singular form (כי ,ותשב ,ותקם, שמעה ,ותצא, 1:6–7), while the daughters-in-law are only mentioned in the circumstantial clause "and her two daughters-in-law" (1:7a).[30] Being the remaining head of an Israelite family, Naomi is socially superior, according to the narrator. Even though she is eventually accompanied by Ruth at her side, Naomi does not acknowledge Ruth's sacrifice. Ruth does not receive any loving words of gratitude or commendation, even though she has forfeited everything, including her faith (1:16–17).[31]

Naomi does not use the word hesed to reflect on Ruth's deeds. She employs the word a second time when she is stunned by the abundance supposedly provided to her by Boaz. Thus, she declares God's hesed as generating Boaz's generosity: "Blessed be he by the Lord, whose kindness (Hebrew חסדו, his hesed) has not forsaken the living or the dead!" (2:20a). Scholars have speculated as to whom the masculine suffix in "his hesed" refers to, God or Boaz (see Collins 1993, p. 100). However, the syntax in the Hebrew seems to be clear,

indicating that the statement refers to God. Namely, God's hesed lies within Boaz's actions. As such, Naomi's statement about hesed resonates with her earlier complaint regarding her calamity, which was also ascribed to God (1:21). Her acknowledgment of God's (and indirectly Boaz's) hesed further highlights her lack of appreciation of Ruth's words and acts of commitment. While Ruth's deeds are recognized by the other protagonists (Boaz and the Bethlehem women, 2:11, 4:15), Naomi remains silent and does not echo the compliments.

Chapter 3 opens with Naomi expressing a wish that Ruth *finds rest* that will do her well. Simply put, Ruth should find a secure home to settle down. In order to do so within the social and cultural framework of the narrative, she must find a husband. Naomi concocts a plan.[32]

The prominent and wealthy kinsman Boaz could marry Ruth and procreate with her. Naomi, therefore, instructs Ruth in detail how to, lightly put, capture Boaz's heart. The nature of Naomi's instructions, whether sexual or otherwise, has occupied scholars due to their semantic ambiguities. Nonetheless, it is clear that the goal is to tempt Boaz into taking Ruth as a wife, or at least as a concubine.

Ruth does as she is instructed. The scene looms with sexual overtones, placing the protagonist into "a crucible of moral choice" (Hubbard 1988, p. 196). Her act of seduction, however, of sneaking in and laying down at the feet of a tired and maybe inebriated man, is not her idea (contra gleaning the field, 2:2). It is planned by Naomi, an elderly woman who is familiar with the local social norms. Although ambiguous, Naomi's instructions to Ruth in launching her to the threshing floor reveal that she requires Ruth to entrap Boaz with sexual intercourse.[33] The references to the special clothing, to the night-time and inebriation, and to the lack of recognition (2:2b–4) are reminiscent of events of sexual intercourse, especially through deceit. Naomi's final instruction, " . . . he will tell you what *to do*" (3:4b), indicates that she anticipates that the couple will act and not talk. It is possible that Naomi supposes that a notable man like Boaz cannot show interest in Ruth, a "young Moabite woman" (נערה מאביה, 2:6),[34] unless under some kind of a cloak or compulsion (Nolan-Fewell and Gunn 1988, p. 106).

Naomi not only plans the scene, but also remotely controls the situation. Upon Ruth's return from the threshing floor, Naomi asks מי-את בתי---in a literal translation, "Who are you, my daughter?" (3:16). The question seems odd, as it implies that Naomi cannot recognize Ruth or that she does not expect to see her. The oddness is resolved in 2QRuth, with a "what" question rather than "who" (i.e., מה את בתי) (see Gladson 2012, p. 274), and by Bible translations, such as NRSVue, which rephrases the question as: "How did things go with you, my daughter?" But the words in the MT (מי-את בתי) are a mindful allusion to Boaz's explicit utterance to Ruth at the threshing floor: "Who are you? . . . May you be blessed by the Lord, my daughter" (3:9–10). Naomi's repetition of Boaz's words suggest that she knows of what has transpired during the night, as if she was there with the couple (see Pardes 2022, p. 32), or, at least, this is the author's way of insinuating this.

However, Ruth's response to Naomi attests to Ruth's own shrewdness. She shares with Naomi the details of the affairs (3:16b), reacting to Naomi's yearning to be informed. She reports in her own initiative that the provision supplied by Boaz was intended for Naomi: "For he said, 'Do not go back to your mother-in-law empty-handed'" (3:17b). This phrase, which Boaz did not utter, indirectly alludes to Naomi's past distress. In the words of Sasson: "What better way to sway her mother-in-law than to recall at such an auspicious moment a term (i.e., ריקם, 'empty-handed') that Naomi used in her deepest despair (1:21)?" (Sasson 1987, p. 326).

Reverting to the night scene, Ruth manages to prevent a full realization of the seduction. In contrast to Naomi's advice, she does not wait for Boaz's instructions (3:4). When he is a tad confounded, Ruth quickly reveals her identity cautiously ("Ruth, your servant," 3:9),[35] and prompts her wish that he will spread his "cloak over your servant." Ruth does not intend to break her oath towards Naomi, as by marrying Boaz she would retain kinship towards her. Indeed, she plays at the service of Naomi, but she strays from the plan orchestrated by her and takes control over the situation. As such, she not only avoids the

questionable strategy planned by her mother-in-law, but as we shall see, she also refrains from repeating the conduct exhibited by her ancestral mothers. By avoiding the perversion planned for her, she ameliorates her ancestorial legacy.

## 8. Rectifying Past Perversions

The scene of Ruth at the threshing floor reminds the readers of Ruth's primal ancestors—Lot and his daughters, the survivors of the destruction in Sodom, and their incestual acts.[36] Like Lot's daughters, Ruth performs in the dark and takes advantage of a man who is under the influence of wine (Gen 19:33, 35; Ruth 3:7–8). Similarly, the man's lack of awareness allows the women, in our case Ruth, to dictate the scene (Gen 19:33, 35; Ruth 3:8). However, unlike Lot's daughters, Ruth seduces Boaz only up to a certain point, and does not go as far as to steal his sperm. Boaz awakens, becomes aware of the situation, and takes control of it (3:8–13). Moreover, the act occurs in an inhabited place (3:6), where norms are enforced, rather than in a cave, isolated and deprived of human regulations (Gen 19:30). Above all, Ruth and Boaz's interaction is not considered incestuous, meaning it does not violate any social precept or norm. While the event is clandestine, thereby avoiding people's judgment (Ruth 3:3, 14), the readers are given a supposedly objective view into the interaction, and may judge it as rather appropriate, while incorporating a pledge to resolve Ruth's predicament in accord with levirate legislation (3:12–13). Lot's daughters use wine to achieve success in their scheme; Ruth uses hesed (v. 10) to attain her (and Naomi's) goals.

Ruth's performance also differs from the actions of her foremothers, who are known for inciting Israel in the desert. The narrative in Numbers recounts the women's sexual temptation with the Israelites who took part in the worship of the women's god, Baal Peor (Num 25:1–3). Ruth's temptation, in comparison, does not threaten Boaz's integrity, since she had already embraced the Israelites and their god. This, in a canonical reading, would supposedly explain and justify the acceptance of the Moabite woman into the nation, despite the known Deuteronomic prohibition (Deut 23:4–6).

The scene at the threshing floor also corrects another supposedly inappropriate act of a foremother, this time of Boaz's lineage—Tamar, who conceived by deceiving her father-in-law, Judah. The "sperm theft" in Tamar's case was done through ruse and disguise (Gen 38:16), followed by a public unveiling of the man's identity, thereby openly indicating his liability (vv. 25–26a). Unlike Tamar, Ruth manages to conceive without the use of deception, and through an uncovered and declared identity.[37]

These three comparisons draw an appealing depiction of Ruth, as not only a conductor of hesed, but also an individual who acts consciously and ethically for the sake of herself and of others. This tendency, however, must be observed within the restricted and limited framework of the society in which she lives, whereas Ruth, like other women in her situation, does not have full agency over her situation.

## 9. Acting within a Social and Cultural Framework

Ruth is a foreigner, with no familial connections in Judah. She is also a widow, lacking a financial safety net. The narrative framework indicates that a woman who is not a member of a clan and a household has less prospects of living a secure and sound life (see also Adams 2014, pp. 41–81). Naomi is indeed a native Judean, but as an elderly widow who cannot have children anymore, she has little chance of remarrying and being financially supported.

Naomi is limited to the conventional dominant patriarchal society surrounding her. She struggles to restore her prominence within this society, adhering to its fundamental value system and applying its conventions to others.[38] Her actions throughout the story are dictated by the hope for a male support, a mandate in the Judean society. However, instead of directly approaching Boaz, she uses her Moabite daughter-in-law to reach this goal.[39] Thus, she remains distant, exposing Ruth to risk. The text is clear about the dangers of entering fields of harvest (2:9),[40] and indicates Naomi's awareness of these risks (2:22), revealing that she does not warn Ruth prior to her dispatch (2:2b).[41]

Naomi's role in initiating and planning the seductive scenario (3:3–4) is another moment of risking Ruth's safety. This moment in the story alludes back, again, to the Sodom story; that is, the scene of Lot, still in the city, offering his daughters to the Sodomites as a substitute for the two guests whom the city's residents intended on sexually assaulting (Gen 19:7–8).[42] Naomi, like Lot, pushes Ruth into the hands of a strange man for what is obviously aimed to be a sexual rendezvous. Ruth adheres to Naomi's authority and abides by her instructions (3:5–6).

Another sign of the social constraints is found in Boaz's inquiry into Ruth's identity in Ruth chapter 2. Boaz does not ask of her name, but rather to whom she belongs (2:5).[43] He asks this before knowing that Ruth is a foreigner, revealing that male dominion did not discriminate between nationalities—all women, including female gleaners, must be some male's possession and not a free independent soul. While it is considered a time when every man did "what was right in his own eyes," Ruth's freedom, as a woman, widow, and foreigner, is rather constricted. The subtext insinuates that Ruth's moral choices are determined within a restricted and rigid social structure.

## 10. When and Who, or Simply Why

The tale transpires in the time of the Judges, but the book's climax makes it clear that the text is later to the time of David. For over a century, scholars endeavored to identify the time and author of the book. This venture has failed to produce any persuasive results, although it seems that the majority tend to side with a post-exilic evaluation (for example, Eskenazi and Frymer-Kensky 2011, p. xvi; Fischer 2001, p. 34; Frevel 1992, p. 29). In order to answer the 'who and when,' one must focus on the 'why.'

The question of the purpose of the book has also been widely debated, with answers varying from etiological to theological.[44] An appealing claim is that the book was created as an opposition to the Deuteronomic laws prohibiting integration with Moabites and Ammonites in God's community (Deut 23:4–6), by calling out attention to David's Moabite blood (Braulik 1999, pp. 1–20; Cook 2015, p. 170; Fischer 2001, p. 34; LaCocque 2004, p. 1; Matthews 2004, p. 212). As such, the tale is also a powerful argument against the anti-foreigner climate of Ezra 9–10, which defines Moabite women as unaccepted marital partners, and the rebuke against Moabite women in Neh 13:23–27, reflecting Solomon's marrying foreigners.

While we cannot confirm that the author knew these post-exilic prohibitions, the book serves as evidence to the debate regarding exclusiveness (cf. intermarriage with Moabites, documented in 1 Chr 4:22, 11:46).[45] Lau, who concludes a post-exilic dating for the book of Ruth, suggests that inclusivity was a required quality for communal survival (Lau 2011, pp. 166–67; also see Adams 2018, p. 136). He emphasizes that the narrative addressed the common reader, in contrast to the elitist and institutionalized message of Ezra-Nehemiah (Lau 2011, pp. 184–88). Similarly, Adams claims that the narrative offers a strong argument against the voices at the time of Ezra and Nehemiah, stressing the need for inclusiveness in a clan-based society, focusing on the vulnerability of widows (Adams 2018, p. 138). The tale combats the negative stance against endogamies by portraying Ruth the foreigner as an embodiment of family loyalty (3:10) and feminine nobility (3:11), a matriarch alongside Rachel and Leah (4:11), and moreover, David's great grandmother (4:17–22). The latter is loudly acknowledged, informing that the great king of Israel, David, had Moabite blood flowing through his veins, and not as a result of trickery.

Another basis for the post-exilic proposition focuses on the "levirate marriage" and the ceremony of the sandal (Ruth 4:7), presupposing an era when the custom was incoherent. While the key root יבם appears in the book (1:15), reminiscent of the duty of the brother-in-law, the ceremony in Ruth 4:7 differs from the strict levirate laws of Deut 25:5–10, according to which a childless brother-in-law is obligated to marry his brother's widow (see Fruchtenbaum 2007, p. 276). It would seem that the authors did their best to describe events passed on through generations, although they allowed details to be warped or misinterpreted, as the message of the book was more portentous than the facts. In a

similar manner, for the purpose of this article, whether the text is pre- or post-exilic, the social and cultural norms it reflects remain a reliable means for unfolding the tale's ethical and theological convictions.

The narrative exhibits the hardships endured by widows and foreigners and their efforts to overcome the lack of a functioning household in a clan-based society. Only by understanding the social framework in which widows faced jeopardy can the book be understood.[46] The tale admits that the possibility of escaping rigid social restrictions is limited. Very few people can achieve it, and only if they are willing to act in non-normative and creative ways. In order to breach the walls of social restrictions, one may need to employ trickery and ruse, but also to demonstrate hesed and a willingness to collaborate.

## 11. Conclusions: The Minor Role of Hesed in a Rigid Society

Since the book of Ruth's main objective is the birth of the male heir leading to David (and in later traditions to Jesus), meaning it contains features of the *Birth of a Hero* paradigm (see Brenner 1985, p. 97; Kara-Ivanov Kaniel 2017, p. 8), many tend to regard the tale as more than just etiological and historical, and rather as indicative of moral and theological depth. Nonetheless, the manifestation of hesed, which scholars tend to highlight in the narrative, seems to be overestimated. Examined in the context of the rigid society in which it appears, the limits of hesed are revealed.

The text emphasizes the circumstances of Ruth's actions. Her social disadvantage is reflected in her own words to Boaz: " . . . Why have I found favor in your sight, that you should take notice of me, when I am a foreigner?" (2:10). Ruth's question indicates the downgraded status of migrants and foreigners. The question may imply that as a foreigner, Ruth may experience not only problems of livelihood, but also xenophobia, as enshrined in the Deuteronomic law against intermarriage. Against this background, one should understand Ruth's earlier statement of commitment to Naomi's god (1:16) as a possibly required condition of those asking to be legitimate immigrants.

Boaz's succor rescues Ruth from her weak social status. However, this salvage is a result of Boaz's fondness of Ruth and his financial calculations. It turns out that without "finding favor" in his sight, and maybe also without proclaiming faithfulness to the Israelite god, Ruth would not easily, if at all, have been accepted into the community, and would not escape the destiny of poverty. She could remain a gleaner in the margins of the society for the rest of her life.

Indeed, the story of Ruth offers hesed as an admirable trait. Ruth's hesed brings upon a change in her social status, stimulating a different attitude towards her, despite being a destitute and foreign individual. However, the change does not occur only as a result of carrying out hesed, but also thanks to Ruth's determination, ingenuity, resourcefulness, and personal sacrifice. Her forfeit is not benevolent, but rather a mandatory step in avoiding danger and changing her destiny, as she wishes to live amongst the exclusive Judeans. In a paraphrase on Al Capone's saying, "one can get much further with a kind word and a gun than you can with a kind word alone," Ruth understands that kindness (or hesed) alone is not suffice.

Hesed also does not occur on a wide scale or as part of a social enterprise, it rather goes so far as only to help the protagonist. The book does not provide any substantial or beneficial tools for those who live in the margins of society, including the destitute, who are willing to forfeit their beliefs in exchange for a morsel of bread.[47] As such, the presupposition of the book of Ruth adheres to the non-utopian world imagined in the book of Deuteronomy: "Since there will never cease to be some in need on the earth, I therefore command you, 'Open your hand to the poor and needy neighbor in your land'" (Deut 15:11). The authors of Ruth admit that the poor will remain as such forever. In layman's terms, if Cinderella wished to change her destiny, she would need to search for the prince and lure him, and not wait for institutional assistance for those in deprivation. Ruth's "Cinderella" destiny is an idiosyncratic situation, in which, like in the story of Joseph, Ruth manages to alter her destiny through her special characteristics and skills. This includes offering

herself through sexual intimacy, a requirement from which Joseph manages to escape on his way to recovery (Gen 39:9–20).

Hesed, apparently, does not include an aspiration to fix the world—referred to by the sages as תיקון עולם (world repair). If the authors had any hope of convincing the readers against the existing social rigidness, such as reflected in the Deuteronomic prohibitions or in Ezra and Nehemiah, it seems that they fell short, as the latter have prevailed, and no profound change has been promised or transpired. On the contrary, as rabbinic legislations developed and expanded throughout the centuries, the noose was furthered tightened.

**Author Contributions:** Conceptualization, G.K.; methodology, G.K.; investigation, G.K. and O.M.; resources, G.K. and O.M.; writing—original draft preparation, G.K. and O.M.; writing—review and editing, G.K. and O.M.; supervision, G.K. All authors have read and agreed to the published version of the manuscript.

**Funding:** This research received no external funding.

**Acknowledgments:** We wish to thank Zeke Piestrup for prompting the production of this article by raising reservations about the role of hesed in Ruth. We also thank the three anonymous referees for their helpful comments and insights regarding the manuscript. The argument of this article was initially presented by Gili Kugler in January 2023 at the University of Haifa, in an event in honor of Ilana Pardes's latest monograph (Pardes 2022).

**Conflicts of Interest:** The authors declare no conflict of interest.

## Notes

1   The Moabites are generally depreciated and slandered in the Hebrew Bible. Incidents such as the Moabite women that seduce Israelites to commit harlotry and worship their Gods (Num 25), the Moabite king Balak who hires Balaam to curse Israel (Num 22–24), the wars with the Moabites attested in Judg 3:12–30, 1 Sam 14:47, 2 Sam 8:2, 2 Kings 3 and 13:20, are but a few, perhaps leading to the Deuteronomic prohibition (23:4–7). Even so, there are instances where Moab is depicted positively (Deut 2:9, 2:27–29).

2   For the use of the "Job-like" definition see recently (Pardes 2022, p. 25).

3   Campbell suggests that the author purposely chooses the term יבמה to keep levirate custom in the minds of his audience. This introduces an irony to the story: the term refers to Orpah, but the custom will be later applicable to Ruth (Campbell 1975, p. 73).

4   This statement may possibly refer only to her familial status, as she left Judah as a wife and mother and returned neither.

5   Although both Ruth and Job deal with personal tragedies, the accounts cannot be parralled on the whole. God is indeed blamed in both, but in the widows' circumstances the narrative shows no direct involvement of God. Moreover, in the latter, the protagonists cause a change in their destiny by performing actions, whereas this is not the case in Job.

6   See (Block 1999, p. 605). For further study of hesed see: (Glueck 1967; Sakenfeld 1978). Hesed is mostly discussed as a divine attribute, but humans are also expected to exhibit it. See (Zobel 1982, pp. 48–71; Younger 2002, p. 294).

7   Pirkei Avot 1:2.

8   Midrash Ruth Rabbah, 2:14, translation Hillel ben David, https://www.betemunah.org/supp11-5.html, (viewed 22 February 2023).

9   Song of Songs narrates erotic fantasies of a female speaker; Lamentations is stated by a female lamenter; Esther recounts the story of a female national savior, and Ruth—a female protagonist (cf., the feminine name of Qoheleth, Ecclesiastes' speaker).

10  (Pardes 2022, p. 38). Similarly, Campbell and Hubbard explain that the book of Ruth holds out the practice of hesed as the ideal lifestyle for Israel (Campbell 1975, pp. 29–30; Hubbard 1988, pp. 72–74).

11  Translation here, and if not stated otherwise, from NRSVue. Other English translations tend to state "kindness/kindly" multiple times in the verse (e.g., NIV, NIRV), though the Hebrew Bible states it only once.

12  Cf. Midrash Ruth Zuta 1:8, which teaches that Naomi's "concern" for her daughters-in-law cloaked her desire to protect her own noble self-image in the Judean society, as their existence embarrassed her.

13  Orpah's decision is not reflected upon negatively. She is the paradigm of the sane and reasonable, acting according to the structures and customs of society. It is implied that a young woman in such circumstances was better off returning to her mother's house (see Trible 1978, p. 172).

14  Vv. 16–17 include Ruth's declaration of her intended actions ("I will go . . . I will lodge"), yet in regard to embracing Naomi's people and god no intended actions are recorded. This may be the author's way of hinting to the end of tale, in an almost prophetic statement of Ruth.

15  Or, according to Trible, even stronger than Abraham's, since she stands alone, possesses nothing, is not called upon by God, has no blessing and no one comes to her aid (Trible 1978, p. 173).

[16]　An in-depth discussion on the various conclusions regarding Ruth's conversion is forthcoming in (Jackson Forthcoming).

[17]　Boaz is introduced as "a prominent rich man of the family of Elimelech" (cf., "a man of standing from the clan of Elimelek," NIV; Bush 1996, p. 100). Some suggest that the term implies not only to wealth, but also to strength, with the Hebrew term איש גבור חיל reminiscent of Gideon (Judg 6:12) and Jephtath (Judg 11:1) (see Campbell 1975, p. 90; Hans Kosmala, "gibbor," TDOT 2:373–7). However, there is no indication to Boaz's fighting in battle etc. No doubt Boaz is a wealthy man, owning lands and servants, but his heroism in the story can only be ascribed to his future actions, "rescuing a family and a name from the curse of oblivion" (see Block 1999, p. 651). This attribution is enhanced by recalling that Boaz evidently contributes to David's birth.

[18]　Whether Ruth asks Naomi's permission for the plan or only shares her thoughts with her is debated. Campbell explains that Ruth is determined after sizing up her situation, not asking permission (Campbell 1975, p. 91). Bush objects to this interpretation, seeing her words as a polite request (Bush 1996, p. 102).

[19]　(Sasson 1989, pp. 42–43). See also (Sasson 1987, p. 324). Bush objects to this view, stating that Sasson erroneously interprets the text. Naomi is surprised upon hearing of Boaz's existence, after Ruth returns, hence no plan was concocted (Bush 1996, p. 103).

[20]　Trible states simply "within human luck is divine intentionality" (Trible 1978, p. 176). Bush regards it as a concomitant circumstance (Bush 1996, pp. 104, 106). Campbell claims that few things happen by chance in biblical thought, and rather, it is God's acting the shadows throughout the book (Campbell 1975, p. 112). This view is reflective of most commentators on this subject. See also (Hals 1969, p. 12). Sasson in contrast, views this as Ruth's expeditious success in locating Boaz's field (Sasson 1989, p. 45). Moreover, she knew she would need to wait for the owner (Boaz, 2:7) to grant permission to glean, a request that the overseer could not grant (Sasson 1989, p. 48).

[21]　The Hebrew Bible uses various terms to depict female subordinates נערה ("girl, damsel"), אמה ("maid, handmaid"), and שפחה ("maid, maid-servant"). These words are used interchangeably in relation to Ruth, though one might need to consider variations in their meanings (see Block 1999, p. 665).

[22]　A linguistic comment regarding the NRSVue's translation should be noted here: the adverb "kindly" is used here, thus introducing a formation of the word hesed into the verse, even though it does not appear in the Hebrew text, which entails the term דברת על לב ("spoken upon heart," 2:13). The meaning of the term is ambiguous in the context. It may mean "to console" (e.g., Gen 50:21; Isa 40:2), but also has a sexual implication of persuasion or seduction (e.g., Gen 34:3; Judg 19:3; Hos 2:16). LaCocque explains that the expected term should have been אל לב ("to [the] heart" instead of "upon the heart") (LaCocque 2004, p. 74). Generally speaking, the occurrence of the various forms of "kindness" in the NRSVue and other translations of Ruth is greater than the presence of the term in the MT.

[23]　This view is argued by Sasson (1987, p. 325; 1989, p. 51). Bush argues that Ruth's puzzlement at this familial reception is the author's way of delighting the audience, as they know of Boaz's identity and standing (Bush 1996, p. 129).

[24]　Whereas Boaz is described as איש גבור חיל, like other biblical heroes, Ruth is the only female in the Bible to be designated אשת חיל (3:11), the feminine parallel, "woman of valor".

[25]　He obviously had a name, which Boaz definitely knew, and yet, the narrator chooses not to expose his identity.

[26]　Hubbard argues that God's blessing is a reward to Ruth due to her hesed towards Naomi (Hubbard 1988, p. 267). Nielsen considers the newborn child as a reward to Ruth for her faithfulness (Nielsen 1997, p. 93). Younger sees the birth of the child as the fulfillment of Boaz's promise (Younger 2002, p. 481).

[27]　Fentress-Williams proposes that the subject of this phrase is both Boaz and Obed, since they are of the same family line, thus elevating David's lineage to a national level, not just limited to Bethlehem (Fentress-Williams 2012, p. 120). Matthews argues that God should be the focus of the phrase (Matthews 2004, p. 241). Others find both interpretations possible (for example, see Bush 1996, p. 255; Campbell 1975, p. 164). It is further possible that the reference is neither to God or Boaz, but to the child born of divine intervention.

[28]　This is the only mention of such a phenomenon in the Hebrew Bible. Hubbard suggests that it occurs due to the special circumstances of the story (Hubbard 1988, p. 276). Younger explains that this is not reflective of an actual tradition, and is only the author's literary tool (Younger 2002, p. 483).

[29]　Campbell suggests that it would be better to read that Naomi was Obed's guardian, and not just his nurse (Campbell 1975, p. 165). Fentress-Williams states that the placing of the child against the bosom is indicative of caring and protection, both by men and women in the Bible (Fentress-Williams 2012, p. 123). Contrary to these views, Hubbard sees this motherly action as symbolic of Naomi's adoption of Obed (Hubbard 1988, p. 274). Nolan-Fewell and Gunn understand this to mean that Ruth was only a surrogate (Nolan-Fewell and Gunn 1988, p. 107).

[30]　Bush claims that this is the narrator's way of reflecting upon the daughters-in-law' ambiguity and uncertainty of their intentions (Bush 1996, p. 85).

[31]　Neither does she receive any kind of welcome upon entering Bethlehem. And see Brenner who points out that while Ruth expresses devotion and selfless love towards Naomi, the love is not mutual (Brenner 1985, p. 97). See also (Ben-Naftali 2015, p. 132). Nolan-Fewell and Gunn argue against Trible's suggestion that Naomi is a model of selflessness as her dominant concern is for her daughters-in-law. In their view, Naomi's main concern is Naomi, as Ruth is an inconvenience, and even a menace. Yet, as opportunity knocks, she uses Ruth for her own benefits (Nolan-Fewell and Gunn 1988, pp. 103–7).

[32]     Under the title "Naomi's Clever Plan (3:1–5)," Hubbard explains that Boaz's "earlier kindness toward Ruth sounded the knock of golden opportunity at the widow's door; Naomi intended to answer it without hesitation" (Hubbard 1988, p. 199).

[33]     For a review of the sexual allusions in the passage see (Bush 1996, pp. 152–53).

[34]     Adams suggests that the repeated mention of Ruth's Moabite ancestry underscores the provocation that her full acceptance would raise in some circles (Adams 2018, p. 133). Although the narrative does not speak negatively of Moabites, it does not speak positively either.

[35]     Ruth does not refer to herself as נערה this time, but אמה. Sasson explains that this title ordinarily denotes a woman who can be taken by a freeman as either concubine or wife (Sasson 1987, p. 325).

[36]     For a survey of the striking allusions of Ruth 3 to Gen 19:30–38 see (Schipper 2016, p. 41).

[37]     This presupposition slightly differs from Kara-Ivanov Kaniel's suggestion that the biblical myth of the "mothers of the Messiah" embraces the "perversions" of the mothers that preceded to Ruth: "why did the author of the Book of Ruth choose to situate these heroines together within the blessing of the elders? This choice attests to the author's fondness for these characters, who unify the split between the positive mother figure (or "the virgin"), and the erotic woman, identified with Tamar and Lot's daughters" (Kara-Ivanov Kaniel 2017, p. 4).

[38]     For a survey on the role of the "house of a father" in the Israelite social system see (Bendor 1996).

[39]     Nolan-Fewell and Gunn explain the situational irony: "She owes her restoration . . . to Ruth the Moabite woman . . . who's radical action challenges the male-centered values that permeate both the story and Naomi's worldview" (Nolan-Fewell and Gunn 1988, p. 107).

[40]     The LXX understood Boaz's warning "I have ordered the young men not to bother you (לבלתי נגעך)" as a reference to potential molestation, using the verb ἅπτω (touch or grasp).

[41]     Nolan-Fewell and Gunn suggest that Naomi's concern in 2:22, after hearing Ruth's experience in the field, also attests to Naomi's sensing the possibility of bounty near at hand (Nolan-Fewell and Gunn 1988, p. 105).

[42]     The subsequent event of the daughters' intercourse with their father in the cave can be seen as the authors' act of poetic justice in reponse to Lot's abusive and exploitative parenting.

[43]     Trible explains that Boaz's inquiry was reflective of the culture. Since Ruth had no owner, the overseer had to refer to her through her foreign nationality and her Judean connection (Trible 1978, p. 176). One should ask, however, if we are at a time of hesed, why not simply describe her as "a widow asking to glean"?

[44]     For a comprehensive survey of the suggested purposes of the book see (Huey 1992, pp. 511–12). Also see (de Villiers and le Roux 2016).

[45]     LaCocque accentuates the sociocultural aspects of the narrative, as its subversive agenda includes openness to foreigners and a flexible interpretation of the Torah (LaCocque 2004, pp. 20–21).

[46]     In respect to the risks cf. Deut 16:11–15; 14:27–29; 24:19–21; 25:5–10.

[47]     More specifically, the book indicates that in order to deal with famine and hunger, one needs to flee their country to escape starvation (1:1–2). Moreover, the story teaches that people would be willing to leave everything behind, upon hearing a rumor of plentifulness in foreign lands (1:6).

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
