# Peer review of "Hesed in Ruth: A Frail Moral Tool in an Inflexible Social Structure"

_religions, doi:10.3390/rel14050604_

Round 1

Reviewer 2 Report

This article is well-written. It includes a nice introduction to the main themes of the book of Ruth, especially that of "hesed", but also to the way in which the status of women in early Israelite society is envisioned. Personally, I do not agree with the conclusions, which are not really based on the author's analysis of the text. It sort of looks as if the author had a "theological" agenda, and was looking for a nail to hang it by. 

My main criticism of this article, is its non-engagement with up-to-date critical scholarship on the book of Ruth: when was it written, by whom, for what purposes? The article avoids these issues. This means, that the articles views on the theological message of Ruth are not contextualized. 

Reviewer 3 Report

In footnote 1, the author refers to the dating of the book of Ruth by Ray (2009), without any comment ("Ray suggests dating the sojourn to Moab somewhere during the 18-year Moabite Oppression in the time of the Judges (Judg 3:1214)"). What implications does Ray's dating have for the current study`? Since the dating of the book of Ruth is a contested issue, and the author refers to other datings in e.g. footnotes 36 and 67, this has to be discussed more thoroughly in footnote 1.

It is occasionally unclear what kind of reading the author wants to perform: a historical, a canonical, or something else. The hermeneutical frame of the interpretation should be clarified.

My main concern with this article relates to the paragraph under the headline “Rectifying Past Perversions”. I neither find this headline nor the parallel reading of the story of Ruth and the episode of Sodoma and Gomorra (which is also highlighted in the abstract) convincing.

It might also be added that there are various representations of Moab and Moabites in the Hebrew Bible, in which a negative portrayal of Moab cannot be taken as a template. In the book of Ruth, hÌ£eseḏ seems to be the most prominent characteristic of the Moabites’ behavior towards the Bethlehemites. In 1 Sam 22:3-4, David sends his parents to the king of Moab to secure their safety (cf. Ruth 1:4); while in 17:12, David’s father Jesse is labeled an Ephraimite from Bethlehem of Judah (cf. Ruth 4:22). The Moabite Ithmah is listed among David’s warriors (1 Chron 11:46), while the Judean men Joash and Saraph married Moabite women and returned to Lehem (1 Chron 4:22). Moabites are known for providing food and water for the Israelites during the wilderness period (Deut 2:27-29; Num 21:11-20) as well as for not doing that (Deut 23:4-5). Moab as a group of people that are disenfranchised because of their cultic practices, inhospitality, and hostility (Deut 23:4-7/Num 21-22, 25:1; in Ezra 9; Neh 13: 1-3; 23-27, marriage with Moabite women are particularly singled out as an unwanted practice, including Solomon’s love for foreign–including Moabite–women leading to apostasy.

At a more technical level: Use one uniform system for referring to Hebrew-in the current shape, the article applies different kinds of transliterations as well as Hebrew letters.

Moreover, it should be stated whether the quotations from the Hebrew Bible are the author’s own translations or taken from somewhere else- if the latter, which one should be listed.

Round 2

Reviewer 1 Report

The authors have radically improved their paper. I am grateful for the cover letter detailing their work. Some problems with language remained, and I have suggested quite a few improvements on the version (hopefully) attached. Please let me know if they don't show up. Beyond my suggestions, which they can accept or not, I recommend that the authors conduct a very careful re-reading to make that the paper they publish as good as it can be.  I don't need to see the paper again.  

Author Response

Dear reviewer,

Thank you very much for confirming the article.
We are grateful and quite amazed by the thorough and generous work you did with the manuscript. This has contributed a lot to improving the article.

We would be very happy to thank you personally once we know who you are.

Reviewer 3 Report

Accepted.

Author Response

Dear reviewer, 

Thanks so much for taking the time to read the comments and the improved version, and thanks very much for approving the article. 

Best regards